# Production of a Monoclonal Antibody to the Nucleocapsid Protein of SARS-CoV-2 and Its Application to ELISA-Based Detection Methods with Broad Specificity by Combined Use of Detector Antibodies

**DOI:** 10.3390/v15010028

**Published:** 2022-12-21

**Authors:** Jinsoo Kim, Dongbum Kim, Kyeongbin Baek, Minyoung Kim, Bo Min Kang, Sony Maharjan, Sangkyu Park, Jun-Kyu Choi, Suyeon Kim, Yong Kyun Kim, Man-Seong Park, Younghee Lee, Hyung-Joo Kwon

**Affiliations:** 1Department of Microbiology, College of Medicine, Hallym University, Chuncheon 24252, Republic of Korea; 2Institute of Medical Science, College of Medicine, Hallym University, Chuncheon 24252, Republic of Korea; 3Department of Biochemistry, College of Natural Sciences, Chungbuk National University, Cheongju 28644, Republic of Korea; 4Division of Infectious Diseases, Department of Internal Medicine, Hallym University Sacred Heart Hospital, College of Medicine, Hallym University, Anyang 14068, Republic of Korea; 5Department of Microbiology, College of Medicine, and the Institute for Viral Diseases, Korea University, Seoul 02841, Republic of Korea

**Keywords:** SARS-CoV-2, COVID-19, N protein, monoclonal antibody, ELISA, variants

## Abstract

The coronavirus disease 2019 pandemic, elicited by severe acute respiratory syndrome coronavirus 2 (SARS-CoV-2), is ongoing. Currently accessible antigen-detecting rapid diagnostic tests are limited by their low sensitivity and detection efficacy due to evolution of SARS-CoV-2 variants. Here, we produced and characterized an anti-SARS-CoV-2 nucleocapsid (N) protein-specific monoclonal antibody (mAb), 2A7H9. Monoclonal antibody 2A7H9 and a previously developed mAb, 1G10C4, have different specificities. The 2A7H9 mAb detected the N protein of S clade, delta, iota, and mu but not omicron, whereas the 1G10C4 antibody recognized the N protein of all variants under study. In a sandwich enzyme-linked immunosorbent assay, recombinant N protein bound to the 1G10C4 mAb could be detected by both 1G10C4 and 2A7H9 mAbs. Similarly, N protein bound to the 2A7H9 mAb was detected by both mAbs, confirming the existence of dimeric N protein. While the 1G10C4 mAb detected omicron and mu with higher efficiency than S clade, delta, and iota, the 2A7H9 mAb efficiently detected all the strains except omicron, with higher affinity to S clade and mu than others. Combined use of 1G10C4 and 2A7H9 mAb resulted in the detection of all the strains with considerable sensitivity, suggesting that antibody combinations can improve the simultaneous detection of virus variants. Therefore, our findings provide insights into the development and improvement of diagnostic tools with broader specificity and higher sensitivity to detect rapidly evolving SARS-CoV-2 variants.

## 1. Introduction

Severe acute respiratory syndrome coronavirus 2 (SARS-CoV-2) belongs to the coronavirus family [1]. Morphologically, an enveloped SARS-CoV-2 genome is positive-sense, single-stranded RNA approximately 30 kilobases in size and is similar to other pathogenic betacoronaviruses [2]. Over the past decades, three pathogenic coronavirus outbreaks have occurred. Severe acute respiratory syndrome coronavirus (SARS-CoV) originated from Guangdong province of China, in 2003. Middle east respiratory syndrome coronavirus (MERS-CoV) originated from Kingdom of Saudi Arabia, in 2012. SARS-CoV-2 originated from Wuhan city of China, in 2019, which has resulted in global transmission, bringing unprecedented economic and health crises [3].

Since the start of the coronavirus disease 2019 (COVID-19) pandemic, 663.6 million confirmed cases and more than 6.59 million deaths have been documented world-wide as of 18 November 2022 [4]. Even after more than 3 years since the emergence of the pandemic, most countries are still fighting to control the spread of the disease. Although different types of COVID-19 vaccines have been directed worldwide, low-income countries remain unvaccinated and vulnerable to COVID-19 [5,6,7]. The development of COVID-19 vaccines along with numerous therapeutic molecules has proven to be critical in reducing the risk of hospitalization and death as well as COVID-19 transmission [8,9,10]. The emergence of many SARS-CoV-2 variants of concern such as B.1.1.7 (alpha), B.1.351 (beta), P1 (gamma), B.1.617.2 (delta), and the recent B.1.1.529 (omicron) variant, has enhanced spread and host antibody response escape, resulting in resistance to vaccines and treatment [11]. Therefore, to help control outbreaks, the improvement of rapid, simple, and cost-effective diagnostic testing approaches is urgently needed at a larger scale than before.

The current extensively used diagnostic method for SARS-CoV-2 detection is quantitative real-time polymerase chain reaction (qRT-PCR), which specifically and sensitively detects viral genomic RNA in a nasopharyngeal swab [12]. However, this test is expensive, time-consuming, and requires laboratory processing by skilled operators to perform PCR procedures using sophisticated real-time thermocyclers [12]. Recently, antigen-detecting rapid diagnostic tests (Ag-RDTs), which are rapid, inexpensive, and easy to use, have emerged as an alternative tool for detection of SARS-CoV-2 [13]. However, these tests have much lower sensitivity when determining asymptomatic individuals with a low viral load [14]. In addition, the antigen recognition capacity of Ag-RDTs is hindered due to the upsurge of mutations in the circulating variants of concern [15].

The most important SARS-CoV-2 antigen is the S protein [1]. The S protein is highly immunogenic as it is located on the surface of the protein. The S1 subunit of spike protein contains an immunogenically crucial receptor binding domain (RBD), which is also a key target of neutralizing antibodies [1]. Another main antigen of SARS-CoV-2 is the N protein. The N protein is a fundamental multifunctional viral protein, and possesses two domains, N-terminal domain (NTD) and C-terminal domain (CTD), and a central linker between them. NTD includes a disordered N-terminal arm and an RNA binding domain. CTD is composed of a dimerization domain and a C-terminal tail [16,17]. The CTD of the N protein is known to participate in N protein dimerization and oligomerization [17]. The N protein functions by binding to viral RNA, which is critical for the formation of the viral ribonucleoprotein complex and genome replication. Moreover, the N protein plays an important role in regulation of the host cell cycle and viral pathogenesis, promoting viral production [18,19]. Because of the above-mentioned versatile characteristics of the N protein and its robust production during viral infection, the N protein is considered a notable target for the diagnosis of SARS-CoV-2 [20]. Furthermore, detection of the SARS-CoV-2 N protein is more sensitive than that of the S protein during early infection [21]. However, it is also true that the importance of the N protein itself and its antibodies for diagnostic purpose and immunity evaluation needs some caution. Recent studies concluded that patients with antibodies to the N protein (without antibodies to RBD of the S protein) failed to exhibit neutralizing antibodies. Additionally, the neutralizing capacity was higher in patients with antibodies against RBD of the S protein compared to those with antibodies to the N protein (86% versus 74%) [22].

Recently, we produced a mouse monoclonal antibody (mAb, 1G10C4 clone) targeting SARS-CoV-2 N protein and confirmed its specific binding to the N protein of the S clade, alpha, and beta variants. It had a sensitivity sufficient to recognize the N protein in approximately 300 plaque-forming units (pfu) of SARS-CoV-2 particles [23]. Using this antibody, we also developed an enzyme-linked immunosorbent assay (ELISA)-based system to detect SARS-CoV-2, targeting the interaction between the SARS-CoV-2 Spike CD and the N protein [23].

In this study, another mAb (2A7H9 clone) was generated, targeting a different epitope region of the SARS-CoV-2 N protein. We found that these antibodies had different degrees of specificity toward the N proteins of various SARS-CoV-2 variants. Based on this difference, we developed an ELISA-based detection system with broad specificity and applicable sensitivity using these antibodies in combination.

## 2. Materials and Methods

### 2.1. Cell Culture

Vero E6 cells derived from the African green monkey kidney were purchased from the Korean Cell Line Bank (Seoul, Republic of Korea). The cells were cultured in Dulbecco’s modified Eagle’s medium (DMEM, Thermo Fisher Scientific, Waltham, MA, USA) supplemented with 10% fetal bovine serum (FBS, Thermo Fisher Scientific), 25 mM 4-(2-hydroxyethyl)-1-piperazineethanesulfonic acid (HEPES), 100 U/mL penicillin, and 100 μg/mL streptomycin at 37 °C in a 5% CO_2_ incubator.

### 2.2. Virus Amplification

The viruses used for this study included the SARS-CoV-2 S clade (betaCoV/Korea/KCD03/2020, NCCP43326), delta variant (hCoV-19/Korea/KDCA119861/2021, lineage B.1.617.2, NCCP43390), mu variant (hCoV-19/Korea/KDCA159392/2021, lineage B.1.621, NCCP43407), omicron variant (hCoV-19/Korea/KDCA447321/2021, lineage B.1.1.529, NCCP43408), and iota variant (hCoV-19/Korea/KDCA82438/2021, lineage B.1.526, NCCP 43387). The SARS-CoV-2 S clade, delta, mu, iota and omicron variants were provided by the National Culture Collection for Pathogens (Osong, Republic of Korea). Viruses were amplified as described previously [24,25,26]. In brief, Vero E6 cells (2 × 10^5^ cells/well) were seeded on six-well plates (Corning, NY, USA) and incubated overnight at 37 °C in 5% CO_2_. The cells were washed and infected with each SARS-CoV-2 variant at a multiplicity of infection (MOI) of 0.01 and incubated for 1 h at 37 °C with shaking every 20 min. After incubation, the medium was replaced with 2 mL of DMEM containing 2% FBS and cultured for 72 h. Thereafter, the cell culture supernatants were collected, and the viruses were quantified by plaque assay on Vero E6 cells as described previously [26,27,28]. After determining the pfu number, the virus stocks were stored at −70 °C for future use. SARS-CoV-2 amplification and relevant experimental procedures were performed in biosafety level 3 conditions in the Research Institute of Medical-Bio Convergence of Hallym University, in accordance with the recommendations of the Institutional Biosafety Committee of Hallym University (Permit number Hallym2022-03).

### 2.3. Synthesis, Construction, and Expression of Biotin Peptide-6 × His-Tagged Coronavirus N Proteins

The nucleotide sequences coding for the SARS-CoV-2 N protein (S clade, nucleotide numbers 28274-29530), MERS-CoV N protein (nucleotide numbers 28566-29804), and HCoV-OC43 N protein (nucleotide numbers 29079-30425) were based on the GenBank IDs MN908947.3, KT029139.1, and NC006213.1, respectively. To obtain recombinant biotin peptide and 6×His-tagged coronavirus N proteins (recombinant SARS-CoV-2 N-Bio-His_6_, MERS-CoV N-Bio-His_6_, and HCoV-OC43 N-Bio-His_6_ protein), nucleotide sequences encoding each coronavirus N protein tagged with a biotin peptide and a hexahistidine (6 × His) were synthesized along with restriction enzyme sites (Not I and Kpn I) at the 5′ and 3′ ends, respectively (Bioneer, Daejeon, Republic of Korea), as described previously [23,29]. The synthesized fusion genes were incorporated into a modified pcDNA 3.4 expression vector (Thermo Fisher Scientific) with an IL-2 signal peptide (pcDNA3.4-SARS-CoV-2 N-Bio-His_6_, pcDNA3.4-MERS-CoV N-Bio-His_6_, and pcDNA3.4-HCoV-OC43 N-Bio-His_6_) for mammalian cell expression in 293F cells. Each biotinylated 6 × His-tagged coronavirus N protein (recombinant SARS-CoV-2 N-Bio-His_6_, MERS-CoV N-Bio-His_6_, and HCoV-OC43 N-Bio-His_6_ protein) was produced using a modified expression vector containing *Bir*A, the *Escherichia coli* biotin ligase (Catalog No. 32408; Addgene, Watertown, MA, USA) as described previously [30]. To obtain recombinant proteins without biotinylation (recombinant SARS-CoV-2 N-His_6_, MERS-CoV N-His_6_, and HCoV-OC43 N-His_6_ protein), the N protein was expressed without the *Bir*A vector. Each coronavirus N protein was purified from 293F culture supernatants after 5–7 days of cell culture using Ni-NTA-agarose (Qiagen, Hilden, Germany) chromatography as described previously [23,29].

### 2.4. Mouse Immunization

Six-week-old BALB/c (female, H-2^b^) mice were purchased from Nara Biotech, Inc. (Seoul, Republic of Korea) and maintained in a specific pathogen-free animal facility of the Experimental Animal Center of Hallym University. The animal protocols were approved by the Institutional Animal Care and Use Committee of Hallym University (Permit Number: HallymR12020-26, Hallym2021-12). BALB/c mice were intraperitoneally immunized three times at 14-day intervals with recombinant SARS-CoV-2 N-Bio-His_6_ protein (50 μg) and CpG-DNA (MB-ODN 4531(O), 50 μg) co-encapsulated with the phosphatidyl-β-oleoyl-γ-palmitoyl ethanolamine:cholesterol hemisuccinate (DOPE:CHEMS) complex as previously described [31,32,33].

### 2.5. Production of a Mouse mAb against the SARS-CoV-2 N Protein

Mice immunized with the SARS-CoV-2 N-His_6_ protein were sacrificed, and their splenocytes were prepared for hybridoma cell production. Briefly, the splenocytes were fused with SP2/0 mouse myeloma cells using polyethylene glycol solution (Sigma-Aldrich, St. Louis, MO, USA). The hybridoma cells producing mAbs against SARS-CoV-2 N protein were selected in HAT medium (Sigma-Aldrich) and HT medium (Sigma-Aldrich) according to the standard hybridoma production method as described previously [33]. Hybridoma cells (2A7H9 clone) were injected into the peritoneal cavity of BALB/c mice, and the ascetic fluids were collected and centrifuged. Using Protein A column chromatography, the anti-SARS-CoV-2 N protein mAb (2A7H9 clone) was purified from the ascites and analyzed by sodium dodecyl sulfate-polyacrylamide gel electrophoresis (SDS-PAGE).

### 2.6. Western Blotting

SARS-CoV-2 particles were lysed with a lysis buffer containing 10 mM HEPES, 150 mM NaCl, 5 mM EDTA, 100 mM NaF, 2 mM Na_3_VO_4_, protease inhibitor cocktail, and 1% NP-40. Each virus lysate was loaded on a 4–12% Bis-Tris gradient gel (Thermo Fisher Scientific) and transferred onto a nitrocellulose membrane. The membrane was blocked with 5% skim milk solution in phosphate-buffered saline containing 0.1% Tween-20 (PBST) and incubated with the mouse anti-SARS-CoV-2 N protein mAb (2A7H9 clone or 1G10C4 clone) and then with horseradish peroxidase (HRP)-conjugated donkey anti-mouse IgG secondary antibody (Catalog No. 715-035-150, Jackson ImmunoResearch Laboratories, Inc., West Grove, PA, USA). Enhanced chemiluminescence solution (Thermo Fisher Scientific) was used to detect immune-reactive proteins.

### 2.7. ELISA for Titration and Isotyping of the Anti-SARS-CoV-2 N Protein mAb

Recombinant SARS-CoV-2 N-Bio-His_6_ protein in bicarbonate buffer (0.1 M, pH 9.6), serially diluted (1:3) or at the concentration of 100 ng/well, was used to coat 96-well immunoplates (Thermo Fisher Scientific) overnight at 4 °C. The plates were blocked with PBST containing 1% bovine serum albumin (BSA) for 1 h. After washing with PBST, serially diluted (1:4) ascites fluid, purified mAb (2A7H9 mAb, 1G10C4 mAb), normal mouse IgG (Catalog No. 10400C, Thermo Fisher Scientific), or normal human IgG (Catalog No. P80-205, Bethyl, Waltham, MA, USA) were added, and then the plates were incubated for 2 h, followed by the addition of HRP-conjugated goat anti-mouse IgG (Catalog No. 115-035-003, Jackson ImmunoResearch Laboratories, Inc.) and incubation for 1 h. HRP-conjugated anti-mouse IgG antibodies (each isotype) (Southern Biotech, Birmingham, AL, USA) were used to determine the isotype of the mAb. The protein-antibody interaction was visualized by color development using tetramethyl benzidine (TMB) peroxidase substrate (Kirkegaard and Perry Laboratories, Gaithersburg, MD, USA), and the absorbance was determined at 450 nm using a Multiskan GO ELISA reader (Thermo Fisher Scientific).

### 2.8. Measurement of mAb Binding Affinity by ELISA

The binding affinity of the SARS-CoV-2 N protein-specific antibody in ascites and the purified mAb solution to recombinant SARS-CoV-2 N protein was measured by a standard ELISA as described previously [29]. Briefly, recombinant SARS-CoV-2 N-Bio-His_6_ protein (100 ng/mL) was used to coat 96-well immunoplates overnight, and the plates were blocked with 1% BSA. To measure the binding affinity of the purified mAb, serially diluted (1:3) SARS-CoV-2 N protein-specific mAb (2A7H9) or normal mouse IgG was added to the wells. After incubation for 2 h, the plates were washed with PBST, followed by incubation with HRP-conjugated goat anti-mouse IgG for 1 h. The TMB peroxidase substrate was used for detection, and the absorbance was measured at 450 nm with a Multiskan GO ELISA reader. SigmaPlot (Chicago, IL, USA) was used to determine the EC_50_ value as described previously [34].

### 2.9. Detection of SARS-CoV-2 N Protein in the Culture Supernatants of Infected Cells Using an ELISA

Biotinylation of SARS-CoV-2 N protein-specific mAbs (2A7H9 and 1G10C4) was performed using an EZ-Link^TM^ Sulfo-NHS-Biotinylation Kit (Catalog No. 21425, Thermo Fisher Scientific) according to the manufacturer’s instructions. Ninety-six-well immuno plates were coated with SARS-CoV-2 N protein-specific mAbs (1G10C4 mAb or 2A7H9 mAb, 100 ng/well) overnight at 4 °C and blocked with 1% BSA. Each SARS-CoV-2 variant particle in Vero E6 cell culture supernatants was lysed with a cell lysis buffer. Recombinant SARS-CoV-2 N-His_6_, MERS-CoV N-His_6_, HCoV-OC43 N-His_6_ protein, or lysates of each SARS-CoV-2 variant were serially diluted (1:3) in PBST and added to each well. After incubation for 2 h at room temperature, biotinylated 1G10C4 mAb and/or biotinylated 2A7H9 mAb (100 ng/well) was added to each well, and the plate was then incubated for 2 h at room temperature. After washing three times with PBST, peroxidase-conjugated avidin (Catalog No. S5512, Sigma-Aldrich) was added to each well, and the reaction was developed using the TMB peroxidase substrate. The amount of SARS-CoV-2 N protein in each well was determined by measurement of the absorbance at 450 nm with a Multiskan GO ELISA reader.

## 3. Results

### 3.1. Production and Characterization of the Mouse mAb against the SARS-CoV-2 N Protein

To develop a novel antibody that specifically detects the SARS-CoV-2 N protein, a purified recombinant SARS-CoV-2 N protein-Bio-His_6_ and CpG-DNA co-encapsulated liposome (DOPE: CHEMS) complex mixture was used to intraperitoneally immunize BALB/c mice. The splenocytes were collected, and the 2A7H9 hybridoma clone was selected as described previously [23]. After priming the peritoneum of the mice with pristane, 2A7H9 hybridoma cells were injected into the mouse to obtain enough mAbs from the hybridoma cells for further experiments. Ten days later, the ascitic fluids, rich with mAbs, were harvested, and the titers of antigen-specific mAbs in ascitic fluids were measured using an ELISA. The results show that all the mice successfully produced the mAbs reactive to SARS-CoV-2 N protein (Figure 1A, Appendix A). Purification of the 2A7H9 mAb by Protein A bead chromatography was confirmed by SDS-PAGE under reducing and non-reducing conditions (Figure 1B). Analysis using an antigen-specific ELISA identified the purified mAb as the IgG2a subclass (Figure 1C, Appendix A). Measurement of the binding affinity of the 2A7H9 mAb to recombinant SARS-CoV-2 N-Bio-His_6_ protein yielded an EC_50_ of 0.174 nM (Figure 1D, Appendix A). To further investigate whether the 2A7H9 mAb specifically binds to the N protein in SARS-CoV-2-infected cells, we performed Western blot analysis using virus particles obtained from culture supernatants of Vero E6 cells infected with various SARS-CoV-2 variants (S clade, delta, iota, mu, and omicron). The 2A7H9 mAb detected the N protein of S clade-, delta-, iota-, and mu-infected cell lysates, but not that of omicron-infected cells (Figure 1E), unlike the 1G10C4 mAb that we developed previously [23], which could recognize the N protein in all variants under study (Figure 1F). Therefore, we developed a novel antibody that binds specifically to the N protein of various SARS-CoV-2 variants. However, its specificity is different from that of the previously developed antibody.

### 3.2. Similarity of the Binding Affinity of the 2A7H9 and 1G10C4 mAbs to the SARS-CoV-2 N Protein

To determine whether the 2A7H9 mAb specifically binds the SARS-CoV-2 N protein, an antigen-specific ELISA was performed. Titration curve analysis showed that the 2A7H9 mAb specifically bound to recombinant SARS-CoV-2 N-His_6_ protein in a concentration-dependent manner, exhibiting a pattern similar to that of the 1G10C4 mAb (Figure 2A, Appendix A). We further analyzed the binding affinity of the two mAbs at various concentrations and found that both antibodies recognized the recombinant SARS-CoV-2 N-His_6_ protein with a similar pattern (Figure 2B, Appendix A). In contrast, none of the three control antibodies, mouse IgG2a, normal mouse IgG, and normal human IgG, recognized the recombinant SARS-CoV-2 N-His_6_ protein, further confirming the binding specificity of the mAbs (Figure 2C, Appendix A).

### 3.3. The 2A7H9 mAb Specifically Binds to the N Protein of SARS-CoV-2 but Not to That of MERS-CoV or HCoV-OC43

To determine whether the 2A7H9 and 1G10C4 mAbs bind specifically to the SARS-CoV-2 N protein but not to those of other coronaviruses, a sandwich ELISA was performed using biotinylated 1G10C4 mAb and biotinylated 2A7H9 mAb. Ninety-six-well plates were coated with 100 ng/well of 1G10C4 mAb or 2A7H9 mAb (Figure 3A,B, Appendix A), and serially diluted recombinant SARS-CoV-2 N-His_6_ protein was added, followed by the addition of biotinylated 1G10C4, 2A7H9 or IgG2a mAbs. Our results demonstrated that the N protein bound to coated 1G10C4 mAb could be detected not only by 2A7H9, but also by 1G10C4 itself (Figure 3A). Similarly, N protein bound to 2A7H9 mAb was detected by both 1G10C4 and 2A7H9 mAbs (Figure 3B). Considering that N protein forms dimers as reported previously [17], the capture and detection by the same antibody matched our expectations. However, it is evident that a monomeric form also exists, because binding of the N protein in the ELISA was decreased when the same antibodies were used for coating and capturing compared to when different antibodies were used. Importantly, MERS-CoV-2 N-His_6_ proteins or HCoV-OC43 N-His_6_ proteins were not detected by either of the 1G10C4 or 2A7H9 biotinylated antibodies (Figure 3C,D, Appendix A). These results confirm the specificity of the two mAbs for SARS-CoV-2 N protein rather than for other coronavirus N proteins based on ELISA results.

### 3.4. Detection of SARS-CoV-2 N Protein in the Lysates of SARS-CoV2 Variants

Because the mAbs efficiently detected the recombinant N protein of SARS-CoV S clade in the ELISA (Figure 3), we then investigated whether these mAbs could detect N proteins of SARS-CoV-2 variants using an ELISA. Because the 1G10C4 mAb could detect the N proteins of all SARS-CoV-2 variants under study (Figure 1F), this antibody was used to capture the N proteins from the lysates of these viruses. We found that the 2A7H9 mAb could not detect the N protein of the omicron variant (B.1.1.529) bound to the 1G10C4 mAb (Figure 4A, Appendix A), which is consistent with the Western blotting result (Figure 1E). We compared the amino acid sequence of the omicron N protein with that of other SARS-CoV-2 variants and found a mutation, the deletion of three amino acids (Δ31–33), in the N-terminal arm region of the omicron N protein (Figure 4B). The inability of the 2A7H9 mAb to detect the omicron variant N protein suggests that the deleted region could be part of the epitope targeted by the antibody. To develop a novel detection method with broad specificity, we tested an ELISA detection system using a combination of our antibodies (Figure 5A). First, the viral lysates were added to a 1G10C4 mAb-coated plate, and the N protein was detected using the biotinylated 1G10C4 mAb by itself. A high degree of binding to the omicron and mu variants was observed, whereas binding to the S clade, delta, and iota variants was low (Figure 5B, Appendix A). When the N protein was detected by the biotinylated 2A7H9 mAb, much higher binding to S clade, mu, delta, and iota was observed compared to the detection with 1G10C4 mAb even though there was no binding to omicron (Figure 5C, Appendix A). Finally, a combined use of the biotinylated 1G10C4 mAb and biotinylated 2A7H9 mAb resulted in the detection of all the variants. Furthermore, S clade, delta, and iota variants were detected with higher affinity compared to the biotinylated 1G10C4 mAb alone (Figure 5C). Therefore, a combination of the mAbs for detection may result in broader specificity and/or higher sensitivity toward N proteins of SARS-CoV-2 variants (Figure 5D, Appendix A). Detection of omicron was reduced by combined use of mAbs, probably because 2A7H9 mAb cannot recognize its N protein, and therefore the combination of the antibodies resulted in the dilution effect. In the case of mu, combined detection was unfavorable. It is likely that the two antibodies somehow inhibit each other’s binding to N protein in mu.

## 4. Discussion

The COVID-19 pandemic is ongoing, and the virus is likely to become a common pathogen; therefore, combating this disease will be a constant challenge [35]. Currently, conventional RT-PCR analysis of viral genes remains the most widely used technique for molecular detection of SARS-CoV-2 [36]. Inexpensive, instant, easily accessible Ag-RDTs are becoming a valuable diagnostic tool for the detection of the SARS-CoV-2 N protein, enabling rapid control measures to limit the spread of the virus [37,38]. Ag-RDTs are helpful in coordinating mass gathering events safely with the use of masks, disinfection, and adequate ventilation [39]. To overcome the reduced sensitivity of Ag-RDTs in individuals with low viral load and the emergence of new SARS-CoV-2 variants that unfavorably affect Ag-RDT performance [40,41,42,43], broad-spectrum Ag-RDTs with high sensitivity are urgently needed. In this study, we developed a new approach to detect novel SARS-CoV-2 variants using a combination of high-affinity anti-SARS-CoV-2 N protein-specific mAbs that could be applied to improve the efficacy of Ag-RDTs.

mAbs are effective in the treatment and diagnosis of pathogens and have other biomedical applications. Multiple studies have been used to develop mAbs that recognize and detect different epitopes on the SARS-CoV-2 N protein. Lee et al. reported the development of a rapid biosensor using mAb pairs that bind to conserved epitopes of the SARS-CoV-2 N protein [44]. One study generated anti-N protein mAbs and applied them to latex-based lateral flow immunoassay strips to detect SARS-CoV-2 [45]. Another study reported the generation of mAbs targeting the SARS-CoV-2 N protein and the development of SARS-CoV-2 antigen test strips based on a colloidal gold immunochromatographic strip technique [46]. We also produced mAbs against the SARS-CoV-2 N protein in the previous and present studies. In the present study, we demonstrated a potential diagnostic benefit by pairing the 2A7H9 mAb with a previously developed mAb, 1G10C4, both of which target the N protein of SARS-CoV-2.

We produced and characterized the 2A7H9 mAb and found that it specifically bound to the SARS-CoV-2 N protein but not to the N proteins of other coronaviruses, including MERS-CoV or HCoV-OC43. Furthermore, we found that the specificity of the 2A7H9 mAb toward the N protein of SARS-CoV-2 variants was different from that of the 1G10C4 mAb. The 2A7H9 mAb detected the N proteins of the S clade, delta, iota, and mu variants but not the omicron N protein, whereas the 1G10C4 antibody recognized all N proteins under study. This could have occurred because the epitope targeted by the 2A7H9 mAb may be altered or missing in the omicron variant. An analysis of the amino acid sequence of N proteins of various SARS-CoV2 variants revealed that the most prominent difference was a three-amino acid deletion, Δ31–33, in the NTD region of the N protein of the omicron variant, compared with that of other variants. Thus, the epitope region of the 2A7H9 mAb may target this deleted region. Therefore, the 2A7H9 mAb is limited in application for SARS-CoV-2 detection because the omicron variant is currently the major strain of SARS-CoV-2 and has high transmissibility. Considering that SARS-CoV-2 variants continue to emerge, this result also suggests the need for diverse antibodies for accurate detection of SARS-CoV-2. Nevertheless, dimerization of the SARS-CoV-2 N protein allows for the utilization of various antibody combinations for detection in samples. As a result, a recombinant N protein of SARS-CoV-2 S clade bound to the 1G10C4 mAb could be detected not only by itself, but also by the 2A7H9 mAb. When we detected the N proteins in virus particles of various SARS-CoV-2 strains bound to the 1G10C4 mAb with different detector antibody compositions, the 1G10C4 mAb alone detected the omicron and mu strains with high efficiency, and the 2A7H9 mAb efficiently detected S clade and mu. However, combined use of the 1G10C4 and 2A7H9 mAbs resulted in the detection of all variants with considerable sensitivity. It is likely that the differential binding affinity of detector mAbs to the N proteins of each variant results in differences in their detection sensitivity, and the weak affinity of a detector mAb can be overcome by signal amplification by detection with two antibodies against different epitopes of the N protein. Therefore, combinations of the detector mAbs may result in broader specificity and higher sensitivity toward the N proteins of SARS-CoV-2 variants. Although ELISA is an accurate and quantitative method established in research laboratory, and therefore we can obtain new insights from the results, Ag-RDT is realistic in usual settings, and a new trial has to be finally translated to Ag-RDT for practical use. In this study, we tried to show by ELISA method that a combined use of antibodies can be a strategy to detect viruses more efficiently. Our approach can be applied to Ag-RDT by adopting 1G10C4 mAb as a capture antibody in the immobilized test line, and the mixture of 1G10C4 mAb and 2A7H9 mAb as a detector antibody in the test solution for sample loading. Surely, the process needs optimization for better performance.

## 5. Conclusions

Our study demonstrates the potential use of combinations of anti-SARS-CoV-2 N protein antibodies to detect multiple SARS-CoV-2 variants. Importantly, our study provides insights for the development of rapid antigen testing methods with broader specificity and higher sensitivity in order to detect rapidly evolving SARS-CoV-2 variants. Therefore, future rapid antigen tests for clinical samples using our strategy are warranted.

## Figures and Tables

**Figure 1 viruses-15-00028-f001:**
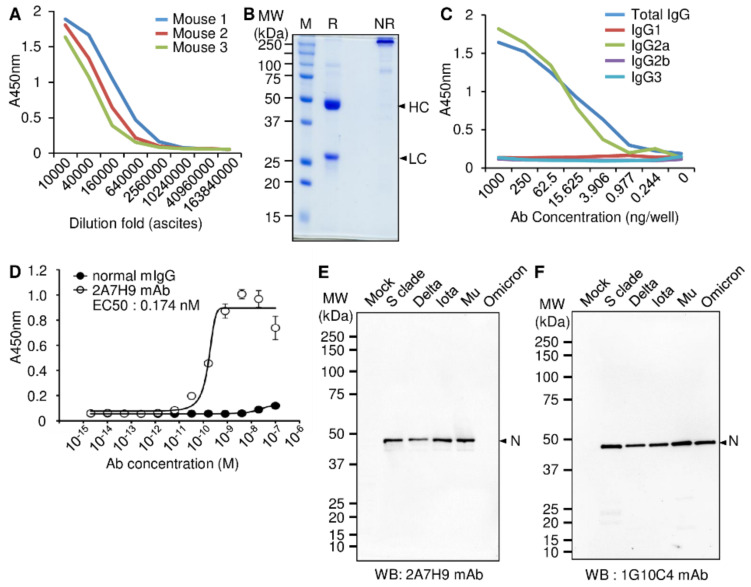
Production and characterization of a mouse mAb against the SARS-CoV-2 N protein. (**A**) Hybridoma cells (2A7H9 clone) were injected into the peritoneal cavity of BALB/c mice (*n* = 3), and ascetic fluid was collected. To estimate the production of SARS-CoV-2 N protein-specific mAbs (2A7H9 mAb) in ascites, an ELISA was performed using a 96-well immuno plate coated with recombinant SARS-CoV-2 N-Bio-His_6_ protein, and a titration curve was obtained. (**B**) The 2A7H9 mAb was purified by Protein A column chromatography and then analyzed by SDS-PAGE in reducing (R) or non-reducing (NR) conditions. HC, heavy chain; LC, light chain. (**C**) The IgG subclass of the purified 2A7H9 mAb was analyzed by an ELISA. (**D**) The binding affinity of the 2A7H9 mAb to recombinant SARS-CoV-2 N protein was determined by an ELISA. (**E**,**F**) Lysates of SARS-CoV-2 S clade, delta, iota, mu, and omicron variants (1 × 10^4^ pfu/well) were separated on a 4–12% Bis-Tris gradient gel. Western blotting was performed with the 2A7H9 mAb (**E**) or 1G10C4 mAb (**F**) to analyze the binding ability of the antibodies to N proteins of SARS-CoV-2 S clade, delta, iota, mu, and omicron variants. N, SARS-CoV-2 N protein.

**Figure 2 viruses-15-00028-f002:**
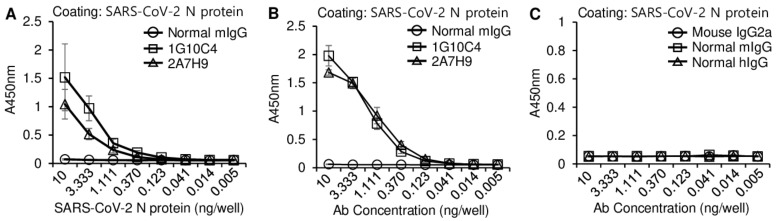
Detection of recombinant SARS-CoV-2 N protein with the 1G10C4 mAb or 2A7H9 mAb. An ELISA was performed to identify binding of the mAbs to recombinant SARS-CoV-2 N protein. (**A**) Ninety-six-well immuno plates were coated with serially diluted (1:3) recombinant SARS-CoV-2 N-Bio-His_6_ protein, and then 100 ng/mL of normal mouse IgG (mIgG), the 1G10C4 mAb or 2A7H9 mAb was added. (**B**) Ninety-six-well immuno plates were coated with recombinant SARS-CoV-2 N-Bio-His_6_ protein (100 ng/well), and serially diluted (1:3) mAbs (2A7H9 or 1G10C4) or normal mIgG was added. (**C**) Recombinant SARS-CoV-2 N-Bio-His_6_ protein (100 ng/well) was used to coat 96-well plates, and serially diluted (1:3) mouse IgG2a, normal mouse IgG (mIgG), or normal human IgG (hIgG) was added. An ELISA was performed to obtain a titration curve.

**Figure 3 viruses-15-00028-f003:**
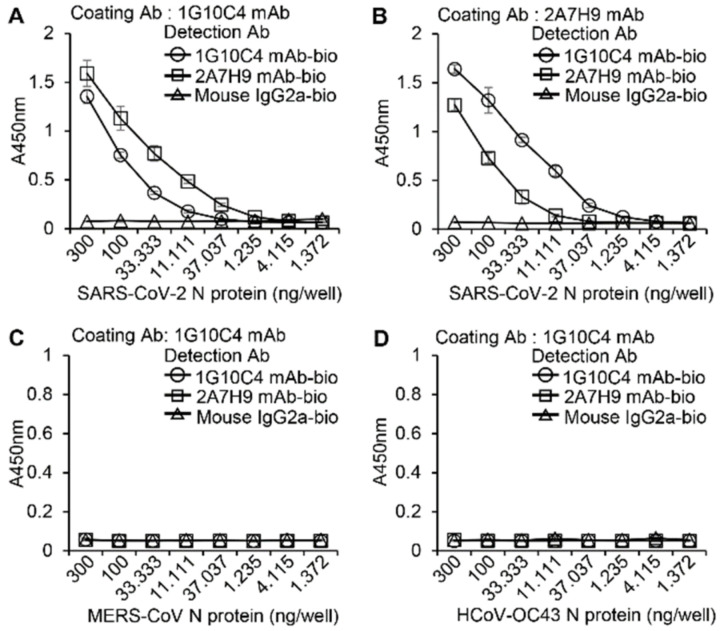
Specificity of the 1G10C4 mAb and 2A7H9 mAb for recombinant SARS-CoV-2 N protein. (**A**,**B**) Ninety-six-well immuno plates were coated with 100 ng/well of 1G10C4 mAb (**A**) or 2A7H9 mAb (**B**), and then serially diluted recombinant SARS-CoV-2 N-His_6_ protein was added. (**C**,**D**) Ninety-six-well immuno plates were coated with 100 ng/well of 1G10C4 mAb before the addition of recombinant MERS-CoV N-His_6_ protein (**C**) or recombinant HCoV-OC43 N-His_6_ protein (**D**). Subsequently, 100 ng/well of the biotinylated 1G10C4 mAb, 2A7H9 mAb, or mouse IgG2a mAb was added, and a titration curve was obtained by an ELISA using peroxidase-conjugated avidin (avidin-HRP).

**Figure 4 viruses-15-00028-f004:**
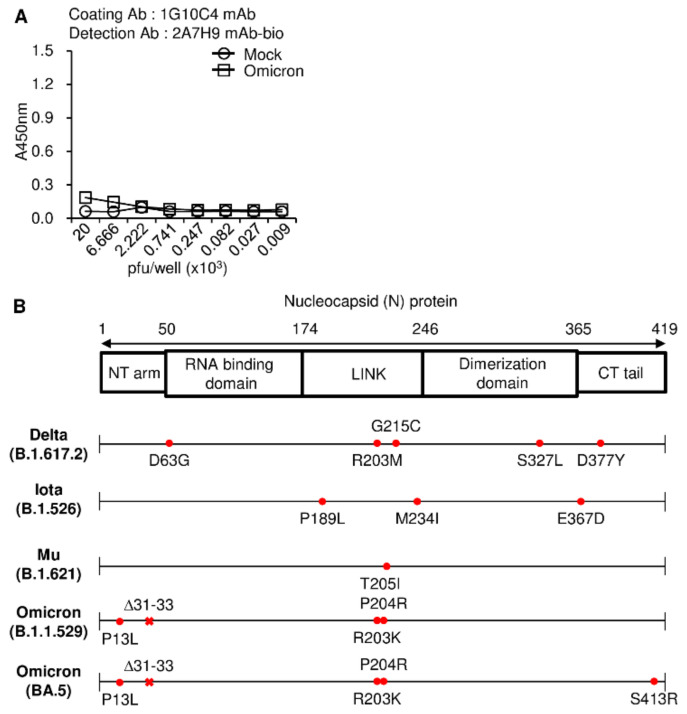
The 2A7H9 mAb does not detect the N protein in virus particles of the SARS-CoV-2 omicron variant. (**A**) Cell culture supernatants containing the SARS-CoV-2 omicron variant were lysed, and serially diluted virus lysates were added to 1G10C4 mAb-coated-immuno plates (100 ng/well) followed by the addition of biotinylated 2A7H9 mAb. Titration was performed after the addition of peroxidase-conjugated avidin (avidin-HRP). (**B**) Analysis of SARS-CoV-2 N protein sequences of SARS-CoV-2 variants. The numbers in the upper row denote the nucleocapsid (N) protein positions in the N gene. The lower rows indicate the mutation residues found in each variant compared to the SARS-CoV-2 S clade sequence. Deletions are indicated by ‘Δ’. NT arm, N-terminal arm; CT tail, C-terminal tail.

**Figure 5 viruses-15-00028-f005:**
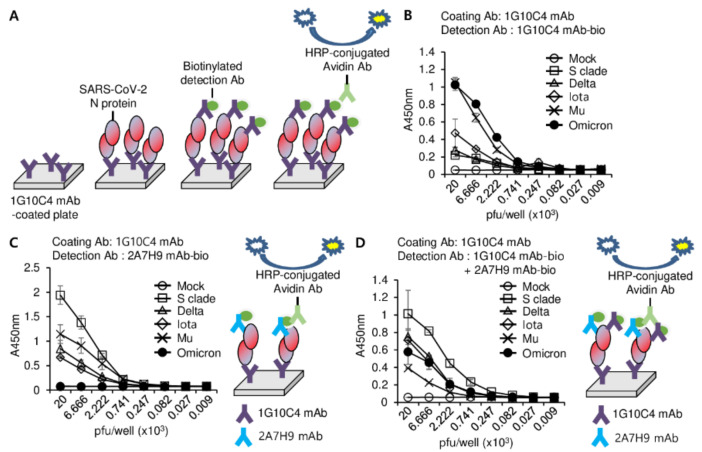
Detection of the N protein in virus particles of SARS-CoV-2 variants using anti-SARS-CoV-2 N protein-specific mAbs. (**A**) Schematic of the detection system. (**B**–**D**) SARS-CoV-2 S clade, delta, iota, mu, and omicron variants in cell culture supernatants were lysed with cell lysis buffer. Ninety-six-well immunoplates were coated with 100 ng/well of 1G10C4 mAb, and then serially diluted lysates of the indicated SARS-CoV-2 variants were added. After washing with PBST, 100 ng/well of biotinylated 1G10C4 mAb (**B**), biotinylated 2A7H9 mAbs (**C**), or a mixture of the biotinylated 1G10C4 and biotinylated 2A7H9 mAbs (**D**) was added, and then the titration curve was obtained by an ELISA using peroxidase-conjugated avidin (avidin-HRP).

## Data Availability

All data needed to evaluate the conclusions in this manuscript have been included.

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
