# Peer review of "Production of a Monoclonal Antibody to the Nucleocapsid Protein of SARS-CoV-2 and Its Application to ELISA-Based Detection Methods with Broad Specificity by Combined Use of Detector Antibodies"

_viruses, 2022, doi:10.3390/v15010028_

Round 1

Reviewer 1 Report

Brief summary

In this study authors have produced monoclonal antibody against N protein of SARS-Cov-2 and characterized it. They have also applied this antibody to develop an ELISA for broad range detection of SARS-Cov-2 variants. Though, manuscript is well written and clearly presents the findings, however, justifying following comments will further strengthen the manuscript –

Comments- 

1.       Line 2 and 23, write N as nucleocapsid in first use.

2.       Line 3, write Elisa as ELISA.

3.       Line 73, “which include an N-terminal arm, a central linker region, and a C-terminal tail”, make a separate line statement to clarify it.

4.       Discuss results of Line 327-330, why omicron and mu binding has reduced in figure 5B compared to 5A.

Author Response

Point-by-point response to the reviewer’s comments

We performed additional experiments and revised our manuscript according to the reviewers’ comments. As a result, we newly inserted the data in Figure 5C and showed working model of the detection system for better understanding in Figure 5. We also provided supplementary information regarding ELISA data. We overall revised the text accordingly with additional introduction and discussion. Now, we believe that our manuscript is much improved.  Thanks for your thoughtful suggestions and critics.

1. Line 2 and 23, write N as nucleocapsid in first use. 

      Response: We revised as you commented. Thanks.

2. Line 3, write Elisa as ELISA.

Response: We wrote as you recommended.

2. Line 73, “which include an N-terminal arm, a central linker region, and a C-terminal tail”, make a separate line statement to clarify it.

Response: We revised this sentence as follows and now the statement is much clearer. Thanks.

“The N protein of SARS-CoV-2 is a fundamental multifunctional viral protein and possesses two domains such as N-terminal domain (NTD) and C-terminal domain (CTD) and a central linker between them. NTD includes a disordered N-terminal arm and an RNA binding domain. CTD is composed of a dimerization domain and a C-terminal tail.” Line77-80. In addition, we revised the description of N protein structure in Figure 4B properly. Thanks.

4. Discuss results of Line 327-330, why omicron and mu binding has reduced in figure 5B compared to 5A.

Response: We believe that it is derived from the difference in the affinity of antibodies toward the N protein of specific strains. To make it clearer, we added the result of the experiments with 2A7H9 mAb-bio as a detection antibody (Fig. 5C). 1G10C4 has higher affinity to N proteins of mu and omicron compared to other strains (Fig. 5B). On the other hand, 2A7H9 has higher affinity to S clade and mu than delta and Iota but does not recognize omicron (Fig. 5C). Through combined use of the two antibodies, we can detect all the variants (Fig. 5D). Decrease of Mu detection capacity by simultaneous treatment with the two antibodies is an interesting feature, and it is likely that somehow the two antibodies inhibit each other to bind to N protein of mu. We revised the part including these results (line344-357) and discussion (Line 424-427). We added schematics for better understanding of the working mechanism. Thanks for your comments.

Reviewer 2 Report

In current study titled ‘Production of a Monoclonal Antibody……’ Kim et al., reported to produce and characterize an anti-SARS-CoV-2 N protein-specific monoclonal antibody (mAb), 2A7H9. The authors present an interesting and important research using sandwich enzyme-linked immunosorbent assay, where recombinant N-protein bound to the 1G10C4 mAb could be detected by both 1G10C4 and 2A7H9 mAbs. 

The author’s need to address following comments:

1.     In introduction section- authors have mentioned that “the detection of the SARS-CoV-2 N protein is more sensitive than that of the S-protein” (line 80-81). Authors need to further elaborate on this point with specific details and references? Spike (S) protein is also highly immunogenic as it is located on the surface of the protein. S1 subunit of spike protein contains immunogenically crucial receptor binding domain (RBD), which is also a key target of neutralizing antibodies. Recent study by MD Anderson lab concluded that patients with antibodies to the N-protein (not S1-RBD) failed to exhibit neutralizing antibodies. Additionally, the neutralizing capacity was higher in patients with antibodies against the S1-RBD compared to N-protein (86% versus 74%) (JCI Insight. 2020;5 (18): e142386. https://doi.org/10.1172/jci.insight.142386).

2.     The authors need to further clarify in detail, at least a theoretical model, as how the generated mAb (2A7H9 clone) provided valuable insights into the improvements of ELISA-based detection, since it was unable to detect the omicron N variants? Thus, limiting its application for SARS-CoV-2 detection, since previously developed mAb 1G10C4 can already recognize the N protein in all variants including omicron N variants.  Also, can the binding affinity of the 2A7H9 mAb to recombinant SARS-CoV-2 N-Bio-His6 also be evaluated through equilibrium dissociation constant (Kd) along with existing EC50?

3.     For reagent dilutions used in ELISA, the authors need to provide data in tabular form – as for the dilution (antigen & positive serum) used for different viral strains.

4.     Three BALB/c mice (female) were injected with 2A7H9 hybridoma cells and ten days later – ascitic fluids rich with mAbs were measured using ELISA; authors need to further discuss and clarify why only 3 mice as one group were selected in the study and not more in different groups?  

5.     In the introduction section- authors’ need to include the details of previous work based on rapid antigen diagnostic tests (Ag-RDT) and how the sandwich enzyme linked assay will fulfill the gaps/lacunae in existing methods? 

6.     authors utilized their existing ELISA technique to measure the binding affinity of the SARS-CoV-2 N protein-specific antibody - all data related to the ELISA assay (ELISA for titration, mAb binding affinity and N protein detection in infected cells) should be included as supplementary tables for better understanding & clarity.

7.     In abstract: “development and improvement of diagnostic tools with broader specificity to de- 33 tect rapidly evolving SARS-CoV-2 variants”. Should it be specificity, or sensitivity, or both?

Author Response

We performed additional experiments and revised our manuscript according to the reviewers’ comments. As a result, we newly inserted the data in Figure 5C and showed working model of the detection system for better understanding in Figure 5. We also provided supplementary information regarding ELISA data. We overall revised the text accordingly with additional introduction and discussion. Now, we believe that our manuscript is much improved.  Thanks for your thoughtful suggestions and critics.

1. In introduction section- authors have mentioned that “the detection of the SARS-CoV-2 N protein is more sensitive than that of the S-protein” (line 80-81). Authors need to further elaborate on this point with specific details and references? Spike (S) protein is also highly immunogenic as it is located on the surface of the protein. S1 subunit of spike protein contains immunogenically crucial receptor binding domain (RBD), which is also a key target of neutralizing antibodies. Recent study by MD Anderson lab concluded that patients with antibodies to the N-protein (not S1-RBD) failed to exhibit neutralizing antibodies. Additionally, the neutralizing capacity was higher in patients with antibodies against the S1-RBD compared to N-protein (86% versus 74%) (JCI Insight. 2020;5 (18): e142386. https://doi.org/10.1172/jci.insight.142386).

Response: Considering the importance S protein and limitation of N protein as a target, we added more information in the first and last parts of the paragraph as follows. As your description was nice, we mostly adopted your sentences and cited the reference properly. Thanks.

“The most important SARS-CoV-2 antigen is the S protein [1]. The S protein is highly immunogenic as it is located on the surface of the protein. The S1 subunit of spike protein contains immunogenically crucial receptor binding domain (RBD), which is also a key target of neutralizing antibodies [1]. Another main antigen of SARS-CoV-2 is the N protein.” Line 73-80.

“However, it is also true that the importance of the N protein itself and its antibodies for diagnostic purpose and immunity evaluation needs some caution. Recent study concluded that patients with anti-bodies to the N protein (without antibodies to RBD of the S protein) failed to exhibit neutralizing antibodies. Additionally, the neutralizing capacity was higher in patients with antibodies against RBD of the S protein compared to those with antibodies to the N protein (86% versus 74%) [22].” Line 88-94.

2. The authors need to further clarify in detail, at least a theoretical model, as how the generated mAb (2A7H9 clone) provided valuable insights into the improvements of ELISA-based detection, since it was unable to detect the omicron N variants? Thus, limiting its application for SARS-CoV-2 detection, since previously developed mAb 1G10C4 can already recognize the N protein in all variants including omicron N variants.  Also, can the binding affinity of the 2A7H9 mAb to recombinant SARS-CoV-2 N-Bio-His6also be evaluated through equilibrium dissociation constant (Kd) along with existing EC50?

Response: Thanks for your thoughtful suggestion. We added working model of ELISA-based detection of SARS-CoV-2 N protein in Fig. 5 (Fig. 5A, C, and D) and inserted new data as Fig. 5C. 1G10C4 mAb can catch all the N proteins in solution and at least one of the two antibodies can detect N proteins of the tested virus strains as N proteins form dimer. Even though 1G10C4 mAb can detect the N protein in all strains, N protein of S clade and delta can be detected with higher sensitivity by 2A7H9 mAb compared to 1G10C4 mAb (Fig. 5B and C). Therefore, the combined use of the two antibodies can improve the sensitivity and specificity against various SARS-CoV-2 variants in general. However, combined detection was unfavorable for mu strain. It is likely that the two antibodies somehow inhibit each other’s binding to N protein in mu.

As far as we understand, the concept of Kd and EC50 is similar. In general, they use SPR (surface plasmon resonance) to accurately measure Kd value. As we don’t have SPR instrument in our institution, it is difficult for us to measure Kd. Please consider this situation.

3. For reagent dilutions used in ELISA, the authors need to provide data in tabular form – as for the dilution (antigen & positive serum) used for different viral strains.

Response: We added all data of ELISA assay in the Supplementary Tables as you instructed.

4. Three BALB/c mice (female) were injected with 2A7H9 hybridoma cells and ten days later – ascitic fluids rich with mAbs were measured using ELISA; authors need to further discuss and clarify why only 3 mice as one group were selected in the study and not more in different groups? 

Response: We screened good candidate among various myeloma cells and a selected superior candidate was the 2A7H9 hybridoma clone. Then, we aimed to obtain enough mAbs from the 2A7H9 hybridoma clone for further experiments. Therefore, we used three mice and found that all the mice successfully produced the antibodies in the ascitic fluid. We are not comparing the group with different groups. We revised to reveal our aim and result of this procedure more clearly.

5. In the introduction section- authors’ need to include the details of previous work based on rapid antigen diagnostic tests (Ag-RDT) and how the sandwich enzyme linked assay will fulfill the gaps/lacunae in existing methods? 

Response: Although ELISA is an accurate and quantitative method established in research lab and therefore we can obtain new insight from the results, Ag-RDT is a realistic way in practical setting and a new trial has to be finally translated to Ag-RDT for practical use. In this study, we tried to show by ELISA that a combined use of antibodies can be a strategy to detect viruses more efficiently. Our approach can be applied to Ag-RDT by adopting 1G10C4 mAb as a capture antibody in the immobilized test line and the mixture of 1G10C4 mAb and 2A7H9 mAb as a detector antibody in the test solution for sample loading. As you know, the procedure needs know-hows and optimization process. Therefore, we added possible application of our ELISA method to Ag-RDT in the last part of discussion as follows.

“Although ELISA is an accurate and quantitative method established in research laboratory and therefore we can obtain new insights from the results, Ag-RDT is realistic in usual setting and a new trial has to be finally translated to Ag-RDT for practical use. In this study, we tried to show by ELISA method that a combined use of antibodies can be a strategy to detect viruses more efficiently. Our approach can be applied to Ag-RDT by adopting 1G10C4 mAb as a capture antibody in the immobilized test line and the mixture of 1G10C4 mAb and 2A7H9 mAb as a detector antibody in the test solution for sample loading. Surely, the process needs optimization for better performance.” Line 432-440.

6. authors utilized their existing ELISA technique to measure the binding affinity of the SARS-CoV-2 N protein-specific antibody - all data related to the ELISA assay (ELISA for titration, mAb binding affinity and N protein detection in infected cells) should be included as supplementary tables for better understanding & clarity.

Response: We added all data of ELISA assay in the Supplementary Tables as you instructed.

7. In abstract: “development and improvement of diagnostic tools with broader specificity to detect rapidly evolving SARS-CoV-2 variants”. Should it be specificity, or sensitivity, or both?

Response: As responded above regarding the improvement of detection method using 2A7H9 mAb, the combined use of the two antibodies as detection antibody has a potential to improve specificity as well as sensitivity. Therefore, we revised the part as follows. Thanks for your valuable question.

Round 2

Reviewer 2 Report

I am satisfied for the revision, which addressed most of my questions.